# Hierarchical Prototypes for Unsupervised Dynamics Generalization in Model-Based Reinforcement Learning

## Abstract

Generalization remains a central challenge in model-based reinforcement learning. Recent works attempt to model the environment-specific factor and incorporate it as part of the dynamic prediction to enable generalization to different contexts. By estimating environment-specific factors from historical transitions, earlier research was unable to clearly distinguish environment-specific factors from different environments, resulting in poor performance. To address this issue, we introduce a set of environment prototypes to represent the environmental-specified representation for each environment. By encouraging learned environment-specific factors to resemble their assigned environmental prototypes more closely, the discrimination of factors between different environments will be enhanced. To learn such prototypes in the unsupervised manner, we propose a hierarchical prototypical method which first builds trajectory embeddings according to the trajectory label information, and then hierarchically constructs environmental prototypes from trajectory prototypes sharing similar semantics. Experiments demonstrate that environment-specific factors estimated by our method have superior clustering performance and can improve MBRL's generalisation performance in six environments consistently.

## 1 Introduction

Reinforcement learning (RL) has achieved great success in solving sequential decision-making problems, *e.g.*, board games (Silver et al., 2016; 2017; Schrittwieser et al., 2020), computer games (Mnih et al., 2013; Silver et al., 2018; Vinyals et al., 2019), and robotics (Levine & Abbeel, 2014; Bousmalis et al., 2018), but it still suffers from the low sample efficiency problem, making it challenging to solve real-world problems, especially for those with limited or expensive data (Gottesman et al., 2018; Lu et al., 2018; 2020; Kiran et al., 2020).In contrast, model-based reinforcement learning (MBRL) (Janner et al., 2019; Kaiser et al., 2019; Schrittwieser et al., 2020; Zhang et al., 2019; van Hasselt et al., 2019; Hafner et al., 2019b;a; Lenz et al., 2015) has recently received wider attention, because it explicitly builds a predictive model and can generate samples for learning RL policy to alleviate the sample inefficiency problem.

As a sample-efficient alternative, the model-based RL method derives a policy from the learned environmental dynamics prediction model. Therefore, the dynamics model's prediction accuracy is highly correlated with policy quality (Janner et al., 2019). However, it has been evidenced that the learned dynamics prediction model is not robust to the change of environmental dynamics (Lee et al., 2020; Seo et al., 2020; Guo et al., 2021), and thus the agent in model-based RL algorithms has a poor generalization ability on the environments with different dynamics. Such a vulnerability to the change in environmental dynamics makes model-based RL methods unreliable in real-world applications where the factors that can affect dynamics are partially observed. For example, the friction coefficient of the ground is usually difficult to measure, while the changes in it can largely affect the dynamics when controlling a robot walking on the grounds, leading to the performance degradation of an agent trained by model-based RL methods (Yang et al., 2019; Gu et al., 2017; Nagabandi et al., 2018b).

Recent Studies (Seo et al., 2020; Nagabandi et al., 2018a; Lee et al., 2020; Guo et al., 2021) have demonstrated that incorporating environmental factor $Z$ into dynamics prediction facilitates the generalisation of model-based RL methods to unseen environments. However, environmental factors

are unobservable in the majority of applications; for instance, the friction coefficient is not available for robots. Therefore, estimating semantical meaningful $Z$ for each environments is the first step for generalization of model-based RL. However, it is not easy to implement, because the environment is hard to label. For example, it is impractical to measure the friction coefficient of every road. Without the label information of environments, $Z$s estimated from previous methods (Seo et al., 2020; Nagabandi et al., 2018a; Lee et al., 2020; Guo et al., 2021) cannot form clear clusters for different environments as Figure 3 shows. These entangled $Z$s cannot represent the distinct environmental specific information, and thus may deviate the learned dynamics prediction function from the true one, resulting in the poor generalization ability.

In this paper, we propose a hierarchical prototypical method (HPM) with the objective of learning an environment-specific representation with distinct clusters. By representing environment-specific information semantically meaningfully, HPM learns more generalizable dynamics prediction function. To achieve this, our method propose to construct a set of environmental prototypes to capture environment-specific information for each environment. By enforcing the estimated $\hat{Z}$ to be more similar to its respective environmental prototypes and dissimilar to other prototypes, the estimated $\hat{Z}$s can form compact clusters for the purpose of learning a generalizable dynamics prediction function. Because environmental labels are not available, we cannot construct environmental prototypes directly. To address this issue, we begin by developing easily-learned trajectory prototypes based on the trajectory label. Then, environmental prototypes can be created by merging trajectory prototypes with similar semantics, as suggested by the natural hierarchical relationship between trajectory and environment.

With the built hierarchical prototypical structure, we further propose a prototypical relational loss to learn $Z$ from past transitions. Specifically, we not only aggregate the $\hat{Z}$s with similar causal effects by optimizing the relational loss (Guo et al., 2021) but also aggregate $\hat{Z}$ with its corresponding trajectory and environmental prototypes via the relational loss. In addition, to alleviate the over-penalization of semantically similar prototypes, we propose to penalize prototypes adaptively with the intervention similarity. In the experiments, we evaluate our method on a range of tasks in OpenAI gym (Brockman et al., 2016) and Mujoco (Todorov et al., 2012). The experimental results show that our method can form more clear and tighter clusters for $\hat{Z}$s, and such $\hat{Z}$s can improve the generalization ability of model-based RL methods and achieve state-of-art performance in new environments with different dynamics without any adaptation step.

## 2 RELATED WORK

**Model-based reinforcement learning** With the learned dynamics prediction model, Model-based Reinforcement Learning (MBRL) takes advantage of high data efficiency. The learned prediction model can generate samples for training policy (Du & Narasimhan, 2019; Whitney et al., 2019) or planning ahead in the inference (Atkeson & Santamaria, 1997; Lenz et al., 2015; Tassa et al., 2012). Therefore, the performance of MBRL highly relies on the prediction accuracy of the dynamics predictive model. To improve the predictive model's accuracy of MBRL, several methods were proposed, such as ensemble methods (Chua et al., 2018), latent dynamics model (Hafner et al., 2019b;a; Schrittwieser et al., 2020), and bidirectional prediction (Lai et al., 2020). However, current predictive methods are still hard to generalize well on unseen dynamics, which hinders the application of MBRL methods in the real-world problems.

**Dynamics generalization in model-based reinforcement learning** To adapt the MBRL to unknown dynamics, meta-learning methods (Nagabandi et al., 2018a;b; Sæmundsson et al., 2018) attempted to update model parameters by updating a small number of gradient updates (Finn et al., 2017) or hidden representations of a recurrent model (Doshi-Velez & Konidaris, 2016). Then, using multi-choice learning, (Lee et al., 2020; Seo et al., 2020) attempted to learn a generalised dynamics model by incorporating environmental-specified information or clustering dynamics implicitly, with the goal of adapting any dynamics without training. Through relational learning and causal effect estimation, RIA (Guo et al., 2021) aims to explicitly learn meaningful environmental-specific information. However, the dynamics change learned by RIA still suffer from a high variance issue.

**Prototypical methods** By learning an encoder to embed data in a low-dimensional representation space, prototypical methods gain a set of prototypical embeddings, which are referred to as

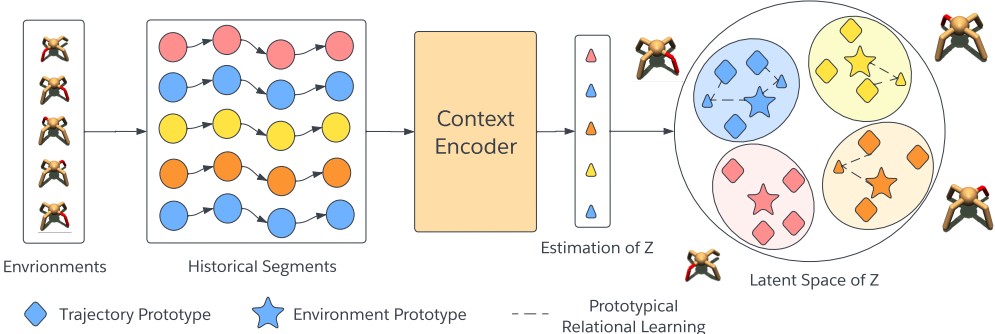

Figure 1: An overview of our Hierarchical Prototypical Method, where the context encoder estimates the environmental-specific factor $\hat{z}$ and environments includes four ants with different destroyed leg with red color. Items extracted from different environments are different colors. We construct prototypes for each trajectory and environment, and denote them as diamond and star, respectively. Each estimated $\hat{z}$ are optimized with its corresponding trajectory and environment prototype using our prototypical relational learning as dotted line shows.

prototypes (Asano et al., 2020; Caron et al., 2020b) that form the basis of this representation space. Prototypical methods aim to derive compact data representations gathering around corresponding prototypes (Li et al., 2021; Oord et al., 2018; Wang et al., 2021), which captures some basic semantic structures. Therefore, prototypical methods have been applied into many areas, *e.g.* self-supervised learning (Li et al., 2020; Caron et al., 2020a), few-shot learning (Snell et al., 2017; Bateni et al., 2020; Simon et al., 2020), domain adaptation (Tanwisuth et al., 2021) and continue learning (De Lange & Tuytelaars, 2021; Yu et al., 2020). In the RL area, (Yarats et al., 2021) ties representation learning with exploration through prototypical representations for image-based RL, while our method focuses on the unsupervised dynamics generalization problem in model-based RL, aiming to learn semantical meaningful dynamics change using prototypical method. Specifically, our method propose a hierarchical method to construct environmental prototypes from trajectory prototypes.

## 3 METHODS

In this section, we first introduce the formulation of the unsupervised dynamic generalization problem in model-based reinforcement learning. Then we present the details of how our hierarchical prototype method learns the environment-specific factors.

### 3.1 PROBLEM SETUP

We formulate the standard reinforcement learning as a markov decision process (MDP) $\mathcal{M} = (\mathcal{S}, \mathcal{A}, r, f, \gamma, \rho_0)$ over discrete time (Puterman, 2014; Sutton & Barto, 2018), where $\mathcal{S}, \mathcal{A}, \gamma \in (0, 1]$ and $\rho_0$ are state space, action space, the reward discount factor, and the initial state distribution, respectively. Dynamics function $f : \mathcal{S} \times \mathcal{A} \to \mathcal{S}$ gives the next state $s_{t+1}$ conditioned on the current state $s_t$ and action $a_t$, and reward function $r : \mathcal{S} \times \mathcal{A} \to \mathbb{R}$ specifies the reward at each timestep $t$ given $s_t$ and $a_t$. The goal of RL is to learn a policy $\pi(\cdot|s)$ mapping from state $s \in \mathcal{S}$ over the action distribution to maximize the cumulative expected return $\mathbb{E}_{s_t \in \mathcal{S}, a_t \in \mathcal{A}}[\sum_{t=0}^{\infty} \gamma^t \, r(s_t, a_t)]$ over timesteps. In model-based RL, we aim to learn a prediction model $\hat{f}$ to approximate the dynamics function $f$, and then $\hat{f}$ can generate training data to train policy $\pi$ or predict the future sequences for planning. With the data provided by learned dynamics model $\hat{f}$, model-based RL has higher data efficiency and better planing ability compared with model-free RL.

In this paper, we consider the unsupervised *dynamics generalization* problem in model-based RL. Different from the standard reinforcement learning, there exists an unobserved variable $Z$ that can affect the dynamics prediction function $f$ in the *dynamics generalization* problem. The goal of *dynamics generalization* is to derive a generalizable policy from given $K$ training MDPs $\{\mathcal{M}_i^{tr}\}_{i=0}^{K}$, and expect the policy can generalize well on $L$ test MDPs $\{\mathcal{M}_j^{te}\}_{j=0}^{L}$. Without losing generality, we assume all MDPs share the same state and action space but preserve different factor $Z$.

In the context of model-based reinforcement learning, we need to learn the dynamics function before learning policy. In order to generalize the dynamic functions on different environment, we need to incorporate the unobserved variable $Z$ into dynamics prediction process, *i.e.*, extending the dynamics function from $f : \mathcal{S} \times \mathcal{A} \to \mathcal{S}$ to $f : \mathcal{S} \times \mathcal{A} \times \mathcal{Z} \to \mathcal{S}$. Since $Z$ is not available, we should estimate it from past transition segments $\tau_{t-k:t-1} = \{(s_{t-k}, a_{t-k}), ..., (s_{t-1}, a_{t-1})\}$ (Seo et al., 2020; Lee et al., 2020; Guo et al., 2021).

Next, we will present how our hierarchical prototypes method estimates $Z$, and enable it to learn the dynamics function $f$ that can generalize to environments with unseen dynamics. In Section 3.2, we present how our method hierarchically constructs prototypes as a representative embedding to represent environmental-specific information for each environment. In Section 3.3, we describe how we update prototypes dynamically and how to estimate environmental-specific factors $Z$ from past transition segments using prototypes. Once $Z$ are estimated, we describe how they enable dynamics function $f$ to generalize well environments with different dynamics.

## 3.2 HIERARCHICAL ENVIRONMENT PROTOTYPES CONSTRUCTION

The objective of our method is to construct a set of prototypes to represent the environmental-specific information for each environment, and guide the context encoder to estimate environmental-specific variable $Z$ from historical transition segments. In each training iteration, we randomly sample a trajectory from a subset of MDPs in the training MDPs. Because labels of MDPs are not available, we cannot estimate environmental prototypes directly. Furtunately, we still have the trajectory label information, and thus we can construct the prototypes for each sampled trajectory first. Specifically, we denote the prototype for $j$-th trajectory as $c_{tra}^{j}$. Because different trajectories may be sampled from a single environment, the trajectory prototypes from the same environment should share similar semantics for dynamics prediction. Therefore, we can construct environmental prototypes hierarchically from trajectory prototypes sharing similar semantics. In this way, environmental prototypes and trajectory prototypes form a natural hierarchical structure, and environmental prototypes can be constructed utilising trajectory label information even if no environmental label is available.

If we denote the $w_{tra}^{i,j}$ as the semantical similarity between the trajectory prototypes $c_{tra}^{i}$ and $c_{tra}^{j}$, we can construct a trajectory similarity matrix $w$ as Figure 2 (b) shows, where each row of $w$, such as $w^{i}$ represents the similarity between $c_{tra}^{i}$ and all other trajectory prototypes. Because it is unknown how many environments are in the sampled trajectories, we directly construct environmental prototypes $c_{env}^{i}$ for each trajectory prototype $c_{tra}^{i}$. Specifically, each environmental prototype $c_{env}^{i}$ is the mean of its corresponding trajectory prototype $c_{tra}^{i}$ and $c_{tra}^{i}$'s top k similar trajectory prototypes.

$$c_{env}^{i} = \frac{1}{K} \sum_{k \in \{T^{i}\}} c_{tra}^{k}, \qquad (1)$$

where $T_i$ denotes the index set of the top-K similar trajectory prototypes with $c_{tra}^{i}$. In this way, we can obtain the $i$-th environmental prototypes, but before that, we need to calculate the semantic similarity matrix $w$.

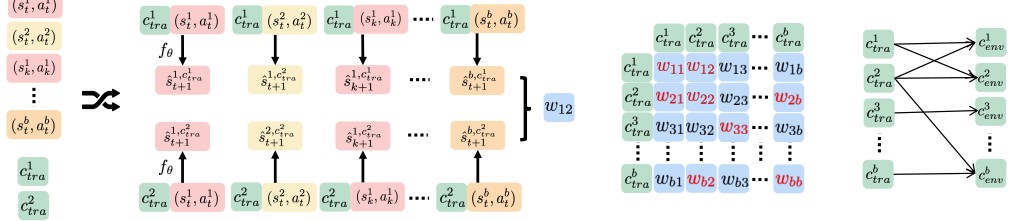

(a) Calculating Similarity between Prototypes via Causal Direct Effect    (b) Merging Top_k Similar Trajectory Prototypes into Environment Prototypes

Figure 2: (a) The illustration of causal direct effect estimation between two trajectory prototypes, where we calculate the mean difference over a batch of predicted next states as the similarity $w$ of the two prototypes. (b) The illustration constructed the similarity matrix between trajectory prototypes. We use the mean of top_k (denoted by the red color) similar trajectory prototypes as the corresponding environmental prototypes.

Normally, we can directly use the euclidean distance to discriminate the similarity between different trajectory prototypes. However, this ignores the semantic effect of trajectory prototypes on dynamics

prediction. If two trajectories prototypes are from a single environment, their trajectory prototypes should share the same semantics, *i.e.*, and their effects on the dynamics function should be the same. Therefore, we consider take account the semantic effect on the dynamics prediction into similarity estimation. However, it is challenging to estimate the effects of the trajectory prototype on the dynamics function because $Z$ is not the only factor that can influence the dynamics function. To remove the effects of other factors, *e.g.* states and actions, on the dynamics function, our method draws inspiration from the recently proposed RIA method (Guo et al., 2021) to calculate the direct causal effects (CDE) of trajectory prototypes. By controlling all factors that have effects on the dynamics function over a mini-batch, we can solely estimate average CDE between different trajectory prototypes as their semantic difference $d$. Concretely, we compute $d$ between two trajectory prototypes using a mini-batch of $S_t$ and $A_t$ pairs $(s_t^i, a_t^i)$ as Figure 2 (a) shows:

$$d_{ij} = \frac{1}{N} \sum_{k=1}^{N} |CDE_{c_{tra}^i, c_{tra}^j}(s_t^k, a_t^k)|, \tag{2}$$

where $N$ is the batch size, $i$ and $j$ are the id of trajectory prototypes. Please refer to Appendix A.6 for the details of CDE. With semantic difference $d$, we can convert it as the semantic difference $w$ via $w = exp(\frac{-d}{\beta})$, where $\beta$ is a factor that controls the sensitivity of $w$. With the calculated similarity $w$, we can construct environmental prototypes via Eq. (1).

Next, we will describe how to update the built trajectory and environmental prototypes to ensure that hierarchical prototypes are representative for each trajectory and environment, and how they help learn the context encoder.

### 3.3 PROTOTYPICAL RELATIONAL LEARNING

As Figure 1 shows, we introduce a context encoder $g$ parameterized by $\phi$ to estimate environmental-specific factor $\hat{z}_t^i$ from the past transition segments $\tau_{t-k:t-1}^i = \{(s_{t-k}, a_{t-k}), ..., (s_{t-1}, a_{t-1})\}$ following previous methods:

$$\hat{z}_t^i = g(\tau_{t-k:t-1}^i; \phi).$$

In order to learn the context encoder and encourage the estimated environmental-specific factor $\hat{z}_t^i$ to be semantically meaningful, we optimize $g$ via the proposed prototypical relational loss to form a clear cluster for $Z$s from the same environments. Concretely, we introduce a relational head (Patacchiola & Storkey, 2020) as a learnable function $h$ to derive the environmental-specific estimation $\hat{z}_t^i$ closely surrounded its associated cluster prototypes. To achieve this, we concatenate the $\hat{z}_t^i$ and its assigned prototypes, *e.g.*, $c_{tra}^i$ as the positive pair, and the concatenation of other prototypes are negative pairs. Then we use the relational head $h$ parameterized by $\varphi$ to quantify the similarity score of $\hat{y}$. To increase the similarity score $\hat{y}$ of positive pairs and decrease those of negatives, we can regard it as a simple binary classification problem to distinguish positive and negative pairs. This can be regarded as maximizing the mutual information between $Z$s and its corresponding prototypes (Please refer to (Tsai et al., 2020; Guo et al., 2021) and Appendix A.3). However, it neglects the semantic correlation among different prototypes, and so it may excessively penalize some semantically relevant prototypes. To alleviate such over-penalization, we propose to penalize prototypes adaptively with the intervention similarity (Guo et al., 2021) through the following objective:

$$\mathcal{L}_{\varphi, \phi}^{i-p-relation} = -\frac{1}{N(N-1)} \sum_{i=1}^{N} \sum_{j=1}^{N} \Big[ [y^{i,j} + (1 - y^{i,j}) \cdot w^{i,j}] \cdot \log h([\hat{z}^i, c^j]; \varphi)$$

$$+ (1 - y^{i,j}) \cdot (1 - w^{i,j}) \cdot \log (1 - h([\hat{z}^i, c^j]; \varphi)) \Big], \tag{3}$$

where $w$ ranges from 0 to 1, and we use it as the similarity between different prototypes. In addition, the first term of Eq. (3) clusters $z_t^i$ with prototypes $c^j$ with the similarity weight $w^{i,j}$, and the second term push them away with weight $1 - w^{i,j}$. To maintain the hierarchical prototypes structure (Li et al., 2020; Guo et al., 2022), we simultaneously update the context encoder by optimizing the objective Eq. (3) between $z$ with trajectory and environmental prototypes. Specifically, the calculation of similarity $w_{env}$ between environmental prototypes and $z$ is same with Section 3.2 as . In addition, we also optimize the relation loss among different $Z$s following (Guo et al., 2021; Li et al., 2020) because $Z$ itself can be regarded as an instance prototype, and thus can retain the property of local smoothness and help bootstrap clustering.

In order to improve its generalization ability on different dynamics, we incorporate the estimated environment-specific $\hat{z}_t$ into the dynamics prediction model $\hat{f}$ and optimize the objective function following (Lee et al., 2020; Seo et al., 2020; Janner et al., 2019):

$$\mathcal{L}_{\theta,\phi}^{pred} = -\frac{1}{N} \sum_{i=1}^{N} \log \, \hat{f}(s_{t+1}^i | s_t^i, a_t^i, g(\tau_{t-k:t-1}^i; \phi); \theta), \tag{4}$$

where $k$ is the length of transition segments, $t$ is the current timestep, and $N$ is the sample size. In addition, we also enable the built prototypes to optimize Eq. (4) to ensure that the learned prototypes are semantically meaningful. Overall, our method simultaneously optimize the prediction loss Eq. (4) and prototypical relational loss Eq. (3) with prototypes in different levels to learn context encoder $g$ and semantic meaningful prototypes, which encourage the estimated environmental-specific $\hat{Z}$ can form clear clusters, and thus can learn a generalizable prediction function $f$.

## 3.4 DIFFERENCE TO RIA

Our paper refers the idea of RIA (Guo et al., 2021) to estimate semantic similarities between different prototypes. However, our method differs RIA from three aspects: 1) RIA estimates the semantic similarities between different instance estimation $\hat{z}$ while our method estimates the semantic similarities between different prototypes. Considering the number of prototypes are limited, the training procedure are faster and more stable than RIA. 2) Our method fully takes the advantage the hierarchy between trajectory and environments, and construct environmental prototype based on trajectory label information while RIA ignores it. Thus our method can achieve better performance than RIA. 3) RIA only pulls $\hat{z}$ and other estimations with similar semantics, but our prototypical relational learning further pulls the $\hat{z}$ and its corresponding trajectory $c_{tra}$ and environmental prototypes $c_{env}$.

## 4 EXPERIMENT

In this section, we perform experiments to evaluate the effectiveness of our approach by answering the following questions: 1) Can our method encourages the learned $Z$ to form a clear cluster? (Section 4.2); 2) Can the learned $\hat{Z}$ with the clear cluster reduce the dynamics prediction errors in model-based RL? (Supplementary Material A.2); 3) Can the learned $\hat{Z}$ with the clear cluster promote the performance of model-based RL in environments with unseen dynamics? (Section 4.3); 4) Is our method sensitive to hyperparameters? (Section 4.4)

### 4.1 ENVIRONMENTAL SETUP

**Implementation details**    Our method includes three learnable functions and a set of learnable trajectory prototypes. The learnable functions are context encoder, relational head and prediction head, and they all are constructed with MLP and optimized by Adam (Kingma & Ba, 2014) with 1e-3 learning rate. During the training procedure, the trajectory segments are randomly sampled from the same trajectory to break the temporal correlations of the training data, which was also adopted by (Seo et al., 2020; Guo et al., 2021). Specifically, we combine $k = 3$ similar trajectory embedding into environmental embedding, and the length of the transition segments is 10, and the hyper-parameters are the same for all experiments, and details can be found in supplementary material A.1.

**Tasks**    Following the previous methods (Lee et al., 2020; Seo et al., 2020), we perform experiments on the classic control algorithm (Pendulum) from OpenAI gym (Brockman et al., 2016) and simulated robotics control tasks (HalfCheetah, Swimmer, Ant, Hopper, Slim-Humanoid) from Mujoco physical engine (Todorov et al., 2012).

**Dynamics settings**    To construct different dynamics of environments, we change the environmental parameters (*e.g.,* length and mass of Pendulum) and predefine them in the training and test environmental parameters lists following previous methods (Zhou et al., 2019; Packer et al., 2019; Lee et al., 2020; Seo et al., 2020; Guo et al., 2021). Specifically, for the convthe training environmental parameters lists for all tasks are $\{0.75, 0.8, 0.85, 0.90, 0.95, 1, 1.05, 1.1, 1.15, 1.2, 1.25\}$, and test environmental parameters lists are $\{0.2, 0.4, 0.5, 0.7, 1.3, 1.5, 1.6, 1.8\}$. We can see that the parameters in test list are out of range of the parameters in the training set. At the training time, we randomly sample the parameters from the training parameter list to train our context encoder and dynamics

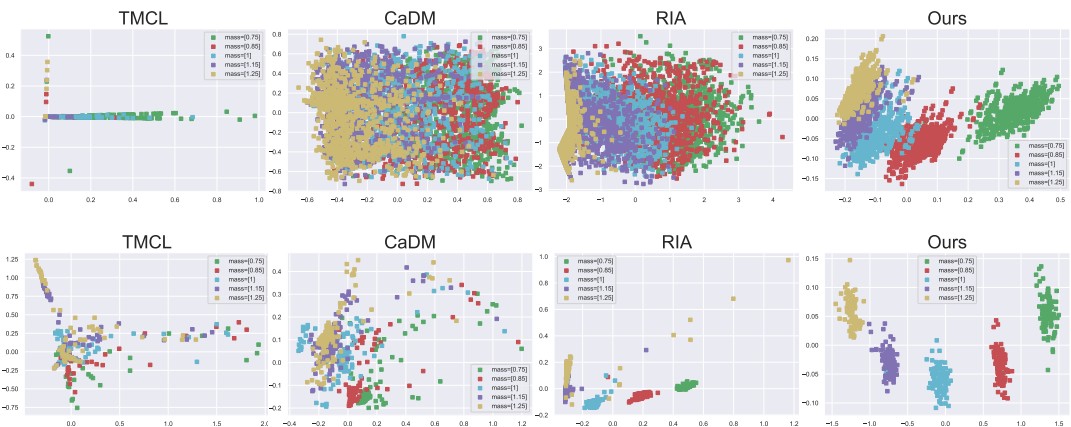

Figure 3: The PCA visualization of environmental-specific factors estimated by TMCL (Seo et al., 2020), CaDM (Lee et al., 2020), RIA (Guo et al., 2021) and ours on the Half-cheetah (upper part) and Pendulum (lower part) task.

prediction model. Then we test our model on the environments with unseen dynamics sampled from the test parameter list. All details are given in supplementary material A.1.

**Planning**    Following (Lee et al., 2020; Seo et al., 2020), we use the model predictive model (MPC) (Maciejowski, 2002) to select actions based on learned dynamics prediction model, and assume that reward functions are known. In addition, we use the cross-entropy method (CEM) (De Boer et al., 2005) to find the best action sequences.

**Baselines**    In this paper, we compare our approach with the following state-of-the-art model-based RL methods on dynamics generalization:

- Context-aware dynamics model (CaDM) (Lee et al., 2020): This method design several auxiliary loss, including backward and future states prediction to learn the context from transition segments.

- Trajectory-wise Multiple Choice Learning (TMCL) (Seo et al., 2020): TMCL introduces multi-choice learning to adapt to different environments. For a fair comparison, we use the no adaptation version of this method.

- Relation Intervention Approach (RIA) (Guo et al., 2021): This method proposes to use relational intervention loss to cluster $Z$s from the same environments.

It has been clearly evidenced that Probabilistic ensemble dynamics model (PETS) (Kurutach et al., 2018) and Meta learning based model-based RL methods, *e.g.* Recurrent model ReBAL and hidden-parameter model GrBAL (Nagabandi et al., 2018b;a), perform worse than CaDM (Lee et al., 2020),TMCL (Seo et al., 2020) and RIA (Guo et al., 2021), so we do not consider them as baselines in our paper.

## 4.2    CLUSTER VISUALIZATION AND ANALYSIS

Table 1: The quantitative evaluation results of estimated environmental-specific factors.

|  | ARI | | | | AMI | | | | V-means | | | |
|---|---|---|---|---|---|---|---|---|---|---|---|---|
|  | TMCL | CaDM | RIA | Ours | TMCL | CaDM | RIA | Ours | TMCL | CaDM | RIA | Ours |
| HalfCheetah | 0.006 | 0.128 | 0.212 | **0.570** | 0.058 | 0.175 | 0.333 | **0.681** | 0.06 | 0.176 | 0.314 | **0.680** |
| Pendulum | 0.060 | 0.471 | 0.754 | **0.971** | 0.054 | 0.529 | 0.838 | **0.967** | 0.051 | 0.531 | 724 | **0.975** |

We perform PCA visualization of estimated $\hat{Z}$s from baselines and our method as Figure 3 to evaluate the cluster performance of estimated $\hat{Z}$s. We can see that our method can achieve better cluster performance qualitatively. Specifically, most $\hat{Z}$s estimated by RIA (Guo et al., 2021) have good cluster performance in general, but the outliers decrease the cluster performance. By contrast, we can

see that there are fewer outliers in our method than them in RIA because the built prototypes and the proposed prototypical relational loss can enforce constraints into estimated $\hat{Z}$s. More qualitatively cluster comparisons can be found in Supplementary Material A.8.

We also quantitatively evaluate the cluster performance of $\hat{Z}$s estimated by baselines and our method. Here we firstly perform k-means (MacQueen et al., 1967) on the estimated $\hat{Z}$s, and then use the ground-truth environmental label to calculate the cluster performance. Here we use the popular mutual information-based metric AMI (Vinh et al., 2010), random-index metric ARI(Hubert & Arabie, 1985) and V-means (Rosenberg & Hirschberg, 2007) as the evaluation metrics. The results are shown in Table 3, we can see that $\hat{Z}$s estimated by our method achieves the highest cluster performance. More quantitatively cluster comparisons can be found in Supplementary Material A.8.

### 4.3 PERFORMANCE COMPARISONS

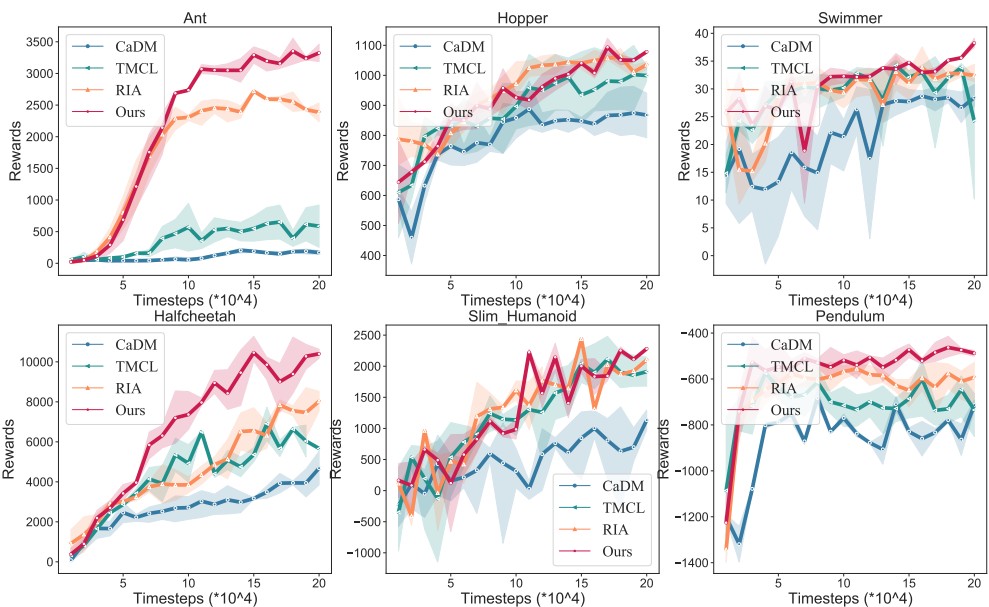

Figure 4: The average returns of model-based RL agents on unseen test environments. The results show the mean and standard deviation of returns averaged over five runs. Specifically, we use the no adaptation version of TMCL for a fair comparison. The performance comparisons on dynamics prediction errors are given at Appendix A.4.

Then, we evaluate the generalization of model-based RL agents trained by our methods and baselines on test environments with unseen dynamics. Following the setting of (Seo et al., 2020), we perform experiments across five runs, and show the test returns on the test environments in Figure 6. Note that the results are slightly different from the results in RIA and TMCL paper since we change the parameter lists that change the environmental dynamics. Specifically, we change the parameter lists of all environments to the same for the convenience of performing environments.

As Figure 6 shows, we can see that our method can achieve significantly better performance than baselines in Ant, Halfcheetah, and Pendulum. Specifically, we can see that our method outperforms the second-best method RIA by 20% in Ant and Halfcheetah environments, which indicates that the changing parameter can largely change their dynamics. In addition, we can see that our method achieves only slightly better performance than baselines in Hopper, Swimmer, and Slim_Humanoid problems. For Hopper and Slim_Humanoid environment, we observe that both RIA and our method can achieve comparable results in all test environments, which indicates that the change of dynamics for Hopper is easy to model and solve. For the Swimmer environment, we observe that TMCL (Seo et al., 2020) sometimes may have a significant performance decline at the final training iteration. This may be because that TMCL may fail to learn the modalities of dynamics function in the no adaptation version. Also, our method still achieves better performance than RIA at the Swimmer task.

## 4.4 ABLATION STUDY

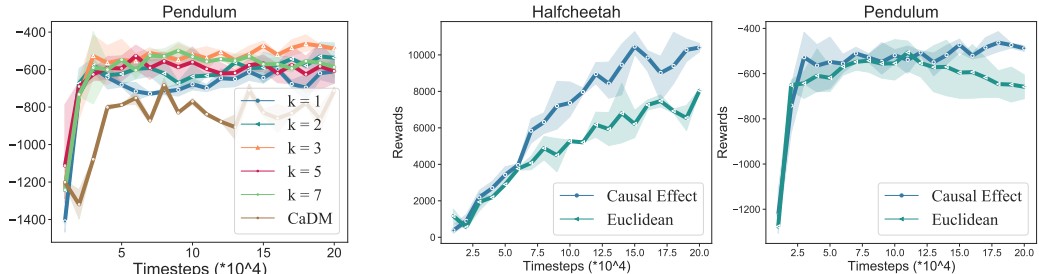

Figure 5: Left Image: The sensitivity analysis about how many trajectory prototypes should be combined into environmental prototypes. Right Two Images: The similarity metrics used in combining trajectory prototypes into environmental prototypes.

In this section, we first perform a sensitive analysis of how many trajectory prototypes should be combined into environmental prototypes. The experiments are conducted at the Pendulum task, and the results are shown as the left image of Figure 5, we can see that no matter what $k$ it is, our method consistently outperforms the baseline CaDM (Lee et al., 2020), which indicates that our method is robust to the selection of $k$ value. Specifically, $k = 1$ means that there are no hierarchical prototypes because one trajectory prototype can decide one environmental prototype, and thus environmental prototypes are the same as trajectory prototypes. We can see that all experimental results with $k > 1$ are better than the experimental result with $k = 1$, which shows the effectiveness of our proposed hierarchical prototypes method and the necessity of the built environmental prototypes. The results of $k = 1$ achieve the best performance on the Pendulum task, so we use it as the default parameter in all experiments.

We also perform an ablation study about the similarity metric used to calculate the similarity among trajectory prototypes. For most cluster methods, *e.g.* k-means (MacQueen et al., 1967), they usually calculate the similarity among entities using the Euclidean distance, while our method uses the direct causal effect as the similarity metric. To evaluate the effectiveness of the similarity metrics based on direct causal effect (Pearl, 2013), we perform experiments on the Halfcheetah and Pendulum tasks, and we can see that using the causal effect to calculate the similarities among trajectory prototypes can achieve better performance than using Euclidean distance on both tasks.

## 5 LIMITATION

Our paper only considers the unsupervised dynamics generalization in model-based reinforcement learning, but model-free RL also suffers from this problem, and we will apply our method to model-free RL in future work. In addition, there are many other generalization problems in reinforcement learning area, *e.g.* observation generalization (Wang et al., 2020; Kirk et al., 2021; Ghosh et al., 2021) and action generalization (Jain et al., 2020), and it would be interesting to extend our method into other generalization settings and train generalizable agents.

## 6 CONCLUSION

In this paper, we focus on the unsupervised dynamics generalization problem in model-based reinforcement learning, and propose a hierarchical prototypical method to construct environmental prototypes in an unsupervised manner. With the learned environmental prototypes, we further propose a prototypical relational loss to learn a context encoder to estimate environmental-specific factors from past transition segments, which enables the dynamics prediction function in model-based reinforcement learning to generalize well on environments with unseen dynamics. The experiments demonstrate that our method can form clearer and tighter clusters for $\hat{Z}$s from the same environment and improve the performance of model-based agents in new environments with unseen dynamics.

## 7 Reproducibility Statement

We acknowledge the importance of reproducibility for research work and try whatever we can to ensure the reproducibility of our work. As for the implementation of our method, details such as hyperparameters are provided in Section 4.1 and Appendix A.1. We will publicly release all codes after the acceptance of this paper.

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

## A  APPENDIX

**We promise that we will public all codes after the acceptance of this paper and we public all
training details at Appendix A.1 and A.3.**

### A.1  ENVIRONMENTAL SETTINGS

We follow the environmental settings of Lee et al. (2020); Guo et al. (2021) and give the details of
settings as follows:

- **Pendulum** We modify the mass $m$ and the length $l$ of Pendulum to change its dynamics.
- **Half-Cheetah** We modify the mass of rigid link $m$ and the damping of joint $d$ of Half-
Cheetah agent to change its dynamics.
- **Swimmer** We modify the mass of rigid link $m$ and the damping of joint $d$ of Swimmer
agent to change its dynamics.
- **Ant** We modify the mass of ant's leg $m$ to change its dynamics. Specifically, we modify
two legs by multiplying its original mass with $m$, and others two with $\frac{1}{m}$.
- **Slim_Humanoid** We modify the mass of rigid link $m$ and the dampling of joint $d$ of the
Slim_Humanoid agent to change its dynamics.
- **Hopper** We modify the mass of $m$ of the Hopper agent to change its dynamics.

Specifically, all training and test parameter lists are set as $\{0.75, 0.8, 0.85, 0.90, 0.95, 1, 1.05, 1.1,
1.15, 1.2, 1.25\}$ and $\{0.2, 0.4, 0.5, 0.7, 1.3, 1.5, 1.6, 1.8\}$, respectively.

### A.2  ALGORITHM

The training procedure is give at Algorithm 1.

### A.3  TRAINING DETAILS

Similar to the Lee et al. (2020); Guo et al. (2021), we train our model-based RL agents and context
encoder for 20 epochs, and we collect 10 trajectories by a MPC controller with 30 horizon from
environments at each epoch. In addition, the cross entropy method (CEM) with 200 candidate actions
is chosen as the planing method. Specifically, the batch size for each experiment is 128, $\beta$ is 6e-1.
All module are learned by a Adam optimizer with 0.001 learning rate.

### A.4  PREDICTION ERROR

### A.5  NETWORK DETAILS

Similar to the Lee et al. (2020), the context encoder is constructed by a simple 3 hidden-layer MLP,
and the output dim of environmental-specific vector $\hat{z}$ is 10. The relational head is modelled as a
single FC layer. The dynamics prediction model is a 4 hidden-layer FC with 200 units.

---

**Algorithm 1** The training algorithm process of our relational intervention approach

---

Initialize parameters of context encoder $\phi$, dynamics prediction model $\theta$ and relational head $\varphi$
Initialize dataset $\mathcal{B} \leftarrow \emptyset$
**for** Each Iteration **do**
    sample environments $\mathcal{M}^i$ from training environments $\{\mathcal{M}^{tr}_i\}^K_{i=0}$          ▷ Collecting Data
    **for** $T = 1$ to TaskHorizon **do**
        Get the estimation of the environment-specified factor $\hat{z}^i_{t-k:t-1} = g(\tau^i_{t-k:t-1}; \phi)$
        Collect $(s_t, a_t, s_{t+1}, r_t, \tau^i_{t-k:t-1})$ from $\mathcal{M}^i$ with dynamics prediction model $\theta$
        Update $\mathcal{B} \leftarrow \mathcal{B} \cup (s_t, a_t, s_{t+1}, r_t, \tau^i_{t-k:t-1})$
        Initialize trajectory prototype $C^i_{tra}$ for each sampled trajectory
    **end for**
    **for** Each Dynamics Training Iteration **do**          ▷ Update $\phi$,$\theta$ and $\varphi$
        **for** $k = 1$ to K **do**
            Sample data $\tau^{i,b,P}_{t-k:t-1}$ , $C^i_{tra}$ and $\tau^{j,b,P}_{t-k:t-1}$ , $C^j_{tra}$ with batch size B,from $\mathcal{B}$
            Get the estimation of the environment-specified factor $\hat{z}^{i,B,P}_{t-k:t-1} = g(\tau^{i,B,P}_{t-k:t-1}; \phi)$ and
            $\hat{z}^{j,B,P}_{t-k:t-1} = g(\tau^{j,B,P}_{t-k:t-1}; \phi)$
            Estimate the similarity $w$ between $C^i_{tra}$ and $C^j_{tra}$
            Construct $w$ between $C^i_{tra}$ and $C^j_{tra}$
            Combing top_k similar $C^i_{tra}$ into environmental prototypes $C^i_{env}$
            $\mathcal{L}^{tot} = \mathcal{L}^{pred}_{\phi,\theta}(\tau^{i,B,,K}_{t:M}, \hat{z}^{i,B,P}_{t-k:t-1}) + \mathcal{L}^{i-relation}_{\phi,\varphi}(\hat{z}^{i,B,P}_{t-k:t-1}, C^i)$ with prototypes in different
levels.
            Update $\theta$ , $\phi$ , $\varphi \leftarrow \nabla_{\theta, \phi, \varphi} \frac{1}{B} \mathcal{L}^{tot}$
        **end for**
    **end for**
**end for**

---

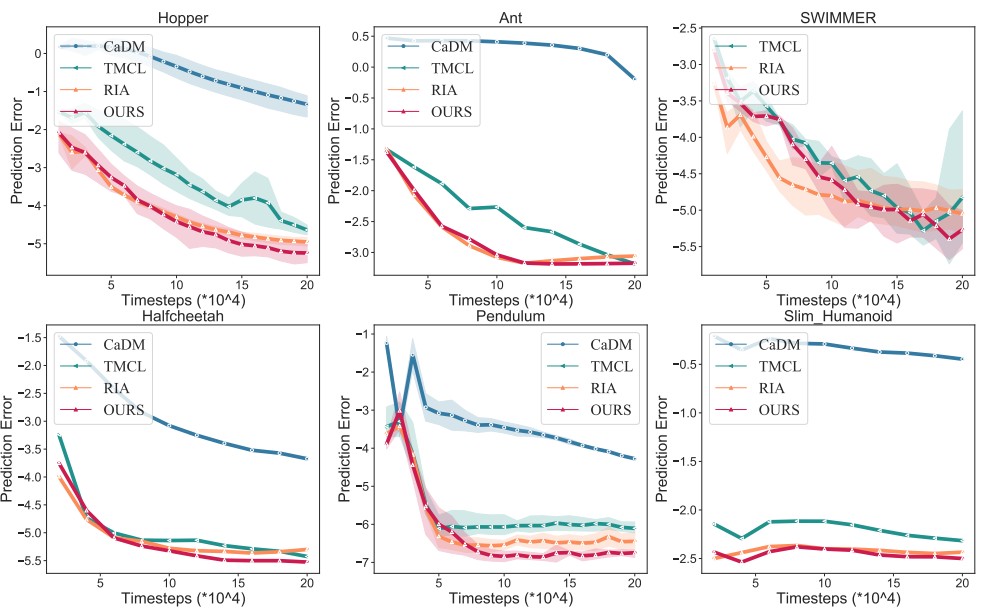

Figure 6: The average prediction errors of model-based RL agents on training environments. The results show the mean and standard deviation of prediction errors averaged over five runs. Specifically, we use the no adaptation version of TMCL for a fair comparison.

Table 2: The average prediction errors of model-based RL agents on test environments. The results show the mean and standard deviation of prediction errors averaged over five runs. Specifically, we use the no adaptation version of TMCL for a fair comparison.

| | TMCL | CaDM | RIA | Ours |
|---|---|---|---|---|
| HalfCheetah | 0.025±0.011 | 0.027±0.013 | 0.021±0.031 | **0.015±0.022** |
| Pendulum | 0.013±0.084 | 0.016±0.004 | 0.003±0.001 | **0.002±0.001** |
| Ant | 0.069±0.013 | 0.054±0.026 | 0.051±0.024 | **0.037±0.012** |
| Hopper | 0.061±0.013 | 0.069 ±0.028 | 0.032±0.023 | **0.026±0.011** |
| Slim_Humanoid | 0.111±0.023 | 0.145±0.021 | 0.105±0.012 | **0.100±0.015** |
| Swimmer | 0.078±0.065 | 0.067±0.085 | 0.037±0.065 | **0.023±0.021** |

## A.6 DIRECT CAUSAL EFFECT BETWEEN TRAJECTORY PROTOTYPES

Concretely, the direct causal effect difference between two trajectory prototypes $c_{tra}^j$ and $c_{tra}^k$ can be calculated through the controllable causal effect (Pearl, 2013) given as following:

$$CDE_{c_{tra}^j, c_{tra}^k}(s_t, a_t) = \mathbb{E}[S_{t+1}|do(S_t = s_t, A_t = a_t), do(Z = c_{tra}^j)] \tag{5}$$

$$- \mathbb{E}[S_{t+1}|do(S_t = s_t, A_t = a_t), do(Z = c_{tra}^k)] \tag{6}$$

$$= \mathbb{E}[S_{t+1}|S_t = s_t, A_t = a_t, Z = c_{tra}^j] - \mathbb{E}[S_{t+1}|S_t = s_t, A_t = a_t, Z = c_{tra}^k], \tag{7}$$

where $do$ is the do-calculus (Pearl, 2000). Because we control all variables that can influence on the $S_{t+1}$, and there is no other confounder between the mediators $(S_t, A_t)$ and $S_{t+1}$ except $Z$ (Guo et al., 2021), we can remove all do operators in Eq. (6). Therefore, the intervention distribution of controlling $Z$, and $(S_t, A_t)$ Eq. (6) is equal to the conditional distribution Eq. (7). In addition, the direct causal effects between $c_{tra}^j$ and $c_{tra}^k$ may differ for different values of $S_t$ and $A_t$, so we should sample $S_t$ and $A_t$ independently of $Z$ to calculate the average controlled direct effect.

Concretely, we directly use a mini-batch of $S_t$ and $A_t$ pairs $(s_t^i, a_t^i)$ to calculate the average controlled direct effect of them as Figure 2 (a) shows:

$$w_{jk} = \frac{1}{N} \sum_{i=1}^N |CDE_{c_{tra}^j, c_{tra}^k}(s_t^i, a_t^i)|, \tag{8}$$

where $N$ is the batch size, $j$ and $k$ are the id of trajectory prototypes.

## A.7 CONNECTION BETWEEN RELATION LOSS AND MUTUAL INFORMATION

We denote the environmental-specific factor as $Z$ and its prototypes as $C$. By definition, the mutual information between $Z$ and $C$ should be:

$$I(Z; C) = \mathbb{E}_{P_{ZC}}[\log(\frac{p(z, c)}{p(z)p(c)})] \tag{9}$$

where $P_{ZC}$ is he joint distribution of $Z$ and $C$, and $P_Z$ and $P_C$ are their marginal distributions. To estimate mutual information between $Z$ and $C$, we can use the probabilistic classifier method proposed by Tsai et al. (2020). Concretely, we can use a Bernoulli random variable $Y$ to classify one given data pair $(z, c)$ from the joint distribution $P_{ZC}$ ($Y = 1$) or from the product of marginal distribution $P(Z)P(C)$ ($Y = 0$). Therefore, the mutual information $I(Z; C)$ between $Z$ and $C$ can be rewrite as:

$$I(Z; C) = \mathbb{E}_{P_{ZC}}[\log(\frac{p(z, c)}{p(z)p(c)})]$$

$$= \mathbb{E}_{P_{ZC}}[\log(\frac{p(z, c|Y = 1)}{p(z, c|Y = 0)})]$$

$$= \mathbb{E}_{P_{ZC}}[\log(\frac{p(Y = 0)P(Y = 1|z, c)}{p(Y = 1)P(Y = 0|z, c)})] \tag{10}$$

Obviously, $\frac{p(Y=0)}{p(Y=1)}$ can be approximated by the sample size, $i.e.$ $\frac{n_{P_Z P_C}}{n_{P_{ZC}}}$, while $\frac{P(Y=1|z,c)}{P(Y=0|z,c)}$ can be measured by a classifier $h(Y|z, c)$ with the below our relational loss:

$$\mathcal{L}_{\varphi,\phi}^{relation} = -\Big[ Y \cdot \log h([z, c]; \varphi) + (1 - Y) \cdot \log (1 - h([z, c]; \varphi)) \Big], \tag{11}$$

where $Y = 1$ if the given pair $(z, c)$ is from the joint distribution $P_{XY}$, and $Y = 0$ if the given pair $(z, c)$ is from the product of the marginal distributions $P_Z P_C$. Because $\frac{p(Y=0)}{p(Y=1)}$ tend to be a constant, optimizing our relation loss is actually estimating the mutual information $I(Z; C)$ between $Z$ and $C$. Therefore, optimizing Eq. (11) is actually maximizing the mutual information between $(\hat{z})$ and its corrsponding prototype which represents the semantics of trajectorey or environment. If the readers are interested in the concrete bound about this method to estimate mutual information, please refer to Tsai et al. (2020); Guo et al. (2021).

## A.8 VISUALIZATION AND ANALYSIS

Table 3: The quantitative evaluation results of estimated environmental-specific factors.

| | ARI | | | | AMI | | | | V-means | | | |
|---|---|---|---|---|---|---|---|---|---|---|---|---|
| | TMCL | CaDM | RIA | Ours | TMCL | CaDM | RIA | Ours | TMCL | CaDM | RIA | Ours |
| HalfCheetah | 0.006 | 0.128 | 0.212 | **0.570** | 0.058 | 0.175 | 0.333 | **0.681** | 0.06 | 0.176 | 0.314 | **0.680** |
| Pendulum | 0.060 | 0.471 | 0.754 | **0.971** | 0.054 | 0.529 | 0.838 | **0.967** | 0.051 | 0.531 | 0.724 | **0.975** |
| Slim_Humanoid | 0.001 | 0.139 | 0.212 | **0.472** | 0.004 | 0.121 | 0.245 | **0.613** | 0.004 | 0.139 | 0.213 | **0.612** |
| Swimmer | 0.052 | 0.583 | 0.586 | **0.615** | 0.052 | 0.582 | 0.597 | **0.638** | 0.012 | 0.528 | 0.595 | **0.637** |

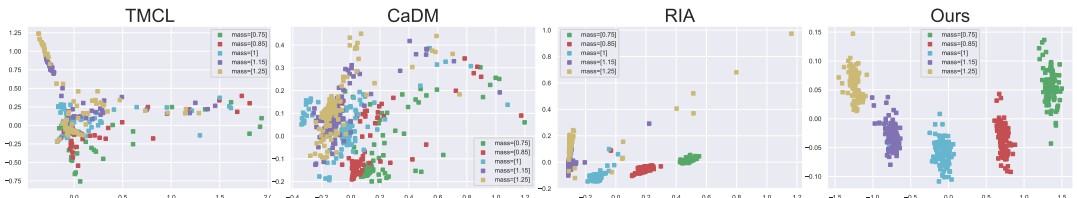

Figure 7: The PCA visualization of environmental-specific factors estimated by TMCL Seo et al. (2020), CaDM Lee et al. (2020), RIA Guo et al. (2021) and ours on the Pendulum task.

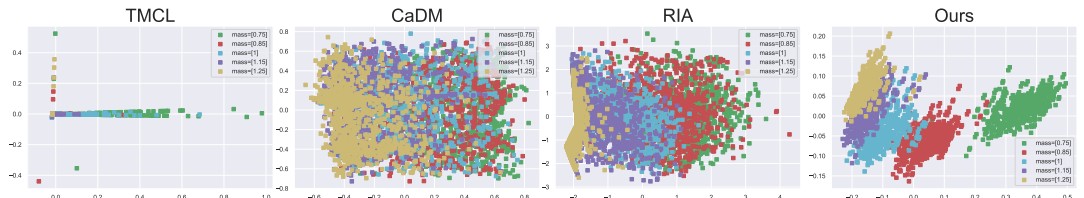

Figure 8: The PCA visualization of environmental-specific factors estimated by TMCL Seo et al. (2020), CaDM Lee et al. (2020), RIA Guo et al. (2021) and ours on the Halfcheetah task.

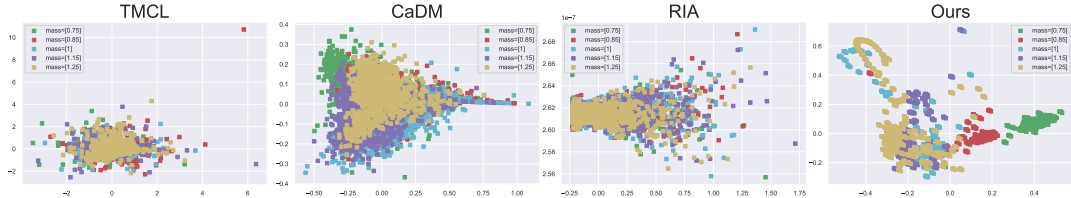

Figure 9: The PCA visualization of environmental-specific factors estimated by TMCL Seo et al. (2020), CaDM Lee et al. (2020), RIA Guo et al. (2021) and ours on the slim_humanoid task.

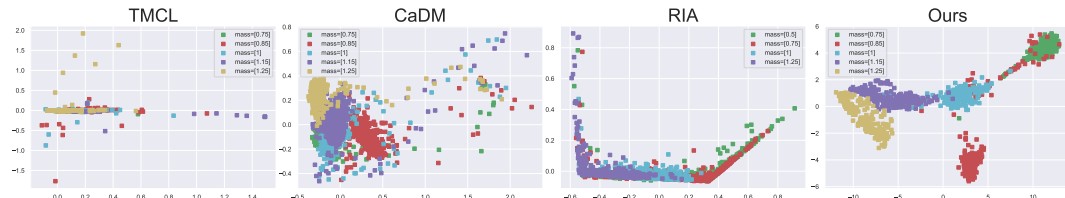

Figure 10: The PCA visualization of environmental-specific factors estimated by TMCL Seo et al. (2020), CaDM Lee et al. (2020), RIA Guo et al. (2021) and ours on the swimmer task.

