# OpenReview forum: "Hierarchical Prototypes for  Unsupervised Dynamics Generalization in Model-Based Reinforcement Learning"
_ICLR.cc/2023/Conference — Submitted to ICLR 2023_

### Official Review · Reviewer_M73e · 2022-10-24

**Confidence:** 4
**Correctness:** 2
**Technical Novelty And Significance:** 1
**Empirical Novelty And Significance:** 2
**Recommendation:** 3

**Clarity, Quality, Novelty And Reproducibility:**

The paper is quite confusing to read, lacks quality and novelty, and is fairly hard to reproduce (given the unclear description).

**Strength And Weaknesses:**


Strengths

- Important and Relevant problem
- Promising general direction for the solution approach

Weakness

- Unclear writing
- Bad paper structure
- Lack of novelty in the proposed approach

**Summary Of The Paper:**

The paper discusses the problem of learning from multiple MDPs, each corresponding to a different value for hidden factors such as physical properties (friction, gravity etc.) The authors propose estimating an embedding based on a similarity matrix of trajectories, and then use a contrastive loss to learn a $z$ that matches with the estimated embeddings. The learnt $z$ is fed back into a dynamics model and used in a MBRL algorithm.

**Summary Of The Review:**

The paper suffers from readability issues, quite critically in parts where the main contribution of the method is introduced. I found myself going back and forth on multiple occasions to try and understand the actual algorithm. From what I could gather, the method is essentially learning an encoding based on how similar trajectories coming from the different MDPs are. I do not see why the problem setup includes using model-based RL as a part of it. That should be part of the solution approach, and not the problem. Similarly, the experimental comparisons are only shown for model-based methods, which does not make sense given the problem setup should have nothing to do with model-based RL in particular. I think the paper needs a heavy revision in terms of the method description, even before going to the experimental results and hence I am advocating for a rejection.

---

### Official Review · Reviewer_wQ75 · 2022-10-24

**Confidence:** 3
**Correctness:** 3
**Technical Novelty And Significance:** 2
**Empirical Novelty And Significance:** 3
**Recommendation:** 5

**Clarity, Quality, Novelty And Reproducibility:**

**Clarity:**
As stated above, clarity is a major issue for this paper.

Some places where the clarity was good:
- The friction coefficient example in the intro was compelling and improved the clarity of the paper.
- the top paragraph of page 4 is really well written and clear.

Some important points that were pretty unclear (in addition to the ones listed previously):
- The abstract is not well written compared to the rest of the intro. One issue is the use of the phrase "the environment-specific factor", which doesn't really make sense. I think the phrase "latent environmental factors" or "unobserved latent environmental factors" would be closer to what you are trying to say.
- Similarly, the intro refers to "environmental factor Z" without introducing Z. This could be improved by referring to latent environmental factors.
- What does it mean to say "we also enable the built prototypes to optimize Eq. 4 to ensure the learned prototypes are semantically meaningful". How are the prototypes used here? What does this accomplish?
- Why is the "no adaptation" version of TCML the more fair comparison? What is it? The results references it repeatedly but the reader has no context on how this affects the method.
- What is the y-axis on Figure 5 (left)?

Some small details to fix:
- Stating "estimating semantically meaningful Z for each environments is **the** first step for generalization of model-based RL" is a bold claim. Maybe say "a promising first step".
- Referring to Figure 3 in the intro without a reference or explanation is disorganized. You can say "our results will show".
- Grammar issues in the intro: "semantically meaningfully", and "our method propose to".
- in the related work, "continue learning" -> "continual learning".
- spelling/grammar issues on p. 5: "i.e., and" (choose either i.e. or and), "consider take account", and "same with Section 3.2 as ."
- p. 6 "convthe"
- p. 7 "model predictive model (MPC)"
- the latex template used for the paper is not the correct version (it should say "Under review as a conference paper at ICLR" on the first page above the top bar)
- p. 9 "sensitive analysis" -> "sensitivity analysis"

**Originality:**
As detailed above, it seems that the most similar work (Guo et al.) also proposed causal and relational learning of environment-specific factors. The difference from Guo et al. is not made clear either in the related work section, or in the methods section. This needs to be improved.

**Quality:**
- It's difficult to assess whether the methods proposed by the paper are based on a reasonable hypothesis about how to estimate environment dynamics, when details like what the CDE (causal direct effects) loss is are never actually explained. Why is CDE a good thing to do? What does it aim to estimate, and how? It also feels disorganized that Figure 2(a) is devoted to illustrating CDE, but a textual explanation of what it does is not provided.
- Similarly, what is the relation loss? Why is it important and a good thing to do?
- The last paragraph of Section 4.1 is a great thing to include (it explains why other methods are not benchmarked against, because they were shown to be worse than the baselines that were included). However, I am curious about the performance of model-free RL in the studied environments. Is model-based RL actually better performing for these tasks?

**Strength And Weaknesses:**

The biggest strength of the paper is that it achieves good results in both clustering and model-based RL performance, as evidenced by Table 1 and Figure 4. To the extent that these results are reproducible, they are of interest to the community.

The biggest issue with the paper is clarity. Highly critical points needed to understand the contributions of the paper and evaluate it are not made clear. For example, Section 3.2 makes conflicting statements about what kind of trajectory and environment labels are available. It first states that no environment labels are available, "so we cannot estimate environmental prototypes directly. Fortunately, we still have the trajectory label information". What is a trajectory label? What information does it provide? Does it indicate in which environment the trajectory was obtained? Wouldn't this be like an environment label? The section then goes on to show that each environment prototype is constructed as the "mean of its corresponding trajectory prototype". How do we know which trajectory prototypes correspond to which environments without environment label information?

The weakness with clarity also impacts the ability to evaluate the novelty of the work. This work seems to be largely based on prior work by Guo et al., including the main losses used (Eq 2 and 3). Therefore, it is not clear what this work contributes beyond the prior work by Guo et al. The authors include a section (3.4) on the "Difference to RIA" (Guo et al.), but this section seems to depend heavily on the point that the proposed approach uses trajectory label information, when (as stated above) it is not clear what a trajectory label *is*. Further, Section 3.4 states that unlike Guo et al., the proposed method estimates the similarity between different prototypes (which I take to mean different trajectories). But, according to the implementation details (Section 4.1), in Guo et al. "the trajectory segments are randomly sampled from the same trajectory", which seems to imply that Guo et al. applies the proposed losses to trajectories as well. Therefore the difference from Guo et al. is very unclear to me. For the authors to improve this point, it would be good to state how their application of the losses proposed by Guo et al. (Eq 2 & 3) differs for their specific use case. This could be done within sections 3.2 and 3.3, rather than having a high-level post-hoc explanation in section 3.4. Evidently there is some difference from RIA, since the results indicate their method can outperform RIA, but how this difference is achieved is not clear.

**Summary Of The Paper:**

The paper is designed to address the issue of model-based reinforcement learning generalizing to new environments with slightly different transition dynamics. Specifically, the case where unobserved environment factors like the friction coefficient of the floor can differ between the training environments and the test environments. The proposed approach estimates latent environmental factors by clustering together trajectories from different environments. The results show that the learned clusters achieve good separation and more closely match the ground truth, and that the technique matches or exceeds the performance of closely related baselines in OpenAI Gym and Mujoco environments.

**Summary Of The Review:**

The clarity issues with the paper make it hard to evaluate the central technical contribution that makes this work different from the closest related prior work (Guo et al.). While the results appear promising, the paper is not yet at the level of an ICLR conference paper.

---

### Official Review · Reviewer_JcMG · 2022-11-02

**Confidence:** 4
**Correctness:** 2
**Technical Novelty And Significance:** 3
**Empirical Novelty And Significance:** 2
**Recommendation:** 5

**Clarity, Quality, Novelty And Reproducibility:**

See above section. Several lines of work are missing in related work, and important baselines have not been considered.

**Strength And Weaknesses:**

*Strengths*
- The proposed clustering method is interesting and intuitive.
- Learning prototypes of environments as a concept is again interesting, intuitive, and to my knowledge novel.
- Proposed method shows stronger results than the baselines considered.

*Weaknesses*
- The authors do not provide a clear motivation for the problem setting. In practice, how is it possible to generate trajectories from a large collection of environments with different factors? What are some practical applications that have these characteristics?
- In my mind, the most plausible way to generate trajectories from diverse environments is through simulation. In this case, there are several sim2real papers that also attempt to learn from simulation and generalize to real world. Such lines of work have not been discussed in related work.
- In Fig 4, what would be the asymptotic performance of an oracle RL agent trained on the unseen test environment. Without this information, it is unclear how good is the performance of proposed method in absolute terms (i.e. relative to an oracle).
- I feel several crucial baselines are missing. Examples are
  - (a) Methods that perform some form of domain randomization, like EPOpt[1], that show that a single policy can be successful despite variations in the environment on several of the tasks considered in this paper.
  - (b) A simple recurrent policy [2] which has been recently shown to be competitive with methods that explicitly perform adaptation.


References
- [1] Rajeswaran et al. 2016. EPOpt: Learning Robust Neural Network Policies Using Model Ensembles.
- [2] Ni et al. 2021. Recurrent Model-Free RL Can Be a Strong Baseline for Many POMDPs.

**Summary Of The Paper:**

This work studies the problem of generalization in (model-based) RL by learning dynamics models that disentangle generalizable factors from environment-specific factors. Specifically, this paper proposes a hierarchical prototypical method (HPM) with the objective of learning to cluster different environments with similar environment-specific factors, thereby facilitating generalization or fast adaptation in new environments by associating it with a learned cluster.

**Summary Of The Review:**

While the proposed algorithmic approach is interesting, the problem setting and assumptions (no simulator; only trajectories from diverse environments) are not well motivated. Important baselines are also missing.

---

### Decision · Program_Chairs · 2023-01-20

**Decision:**

Reject

**Justification For Why Not Higher Score:**

The paper's contributions could not be properly accessed and there was no author response to clarify the misunderstandings.

**Justification For Why Not Lower Score:**

N/A

**Metareview: Summary, Strengths And Weaknesses:**

This paper proposes a new method for learning environment dynamics models that can adapt to different environments. The reviewers point to several issues with clarity that make it difficult to: distinguish how the new method is different from prior work, understand when the technique is applicable, and interpret some of the results. Since there is no author response to address concerns from the reviewers, I cannot recommend this paper for acceptance.

**Summary Of Ac-Reviewer Meeting:**

N/A